# Tuberculosis Severity Predictive Model Using Mtb Variants and Serum Biomarkers in a Colombian Cohort of APTB Patients

**DOI:** 10.3390/biomedicines11123110

**Published:** 2023-11-22

**Authors:** Juan C. Ocampo, Juan F. Alzate, Luis F. Barrera, Andres Baena

**Affiliations:** 1Grupo de Inmunología Celular e Inmunogenética (GICIG), Universidad de Antioquia (UdeA), Medellín 050010, Colombia; juan.ocampom1@udea.edu.co (J.C.O.); luis.barrera@udea.edu.co (L.F.B.); 2Departamento de Microbiología y Parasitología, Facultad de Medicina, Universidad de Antioquia (UdeA), Medellín 050010, Colombia; jfernando.alzate@udea.edu.co; 3Centro Nacional de Secuenciación Genómica (CNSG), Facultad de Medicina, Universidad de Antioquia (UdeA), Medellín 050010, Colombia; 4Instituto de Investigaciones Médicas, Universidad de Antioquia (UdeA), Medellín 050010, Colombia

**Keywords:** *Mycobacterium tuberculosis*, severity model, CHIT1, tuberculosis

## Abstract

Currently, tuberculosis (TB) is a bacterial infection caused by *Mycobacterium tuberculosis* (Mtb) that primarily affects the lungs. The severity of active pulmonary TB (APTB) is an important determinant of transmission, morbidity, mortality, disease experience, and treatment outcomes. Several publications have shown a high prevalence of disabling complications in individuals who have had severe APTB. Furthermore, certain strains of Mtb were associated with more severe disease outcomes. The use of biomarkers to predict severe APTB patients who are candidates for host-directed therapies, due to the high risk of developing post-tuberculous lung disease (PTLD), has not yet been implemented in the management of TB patients. We followed 108 individuals with APTB for 6 months using clinical tools, flow cytometry, and whole-genome sequencing (WGS). The median age of the study population was 26.5 years, and the frequency of women was 53.7%. In this study, we aimed to identify biomarkers that could help us to recognize individuals with APTB and improve our understanding of the immunopathology in these individuals. In this study, we conducted a follow-up on the treatment progress of 121 cases of APTB. The follow-up process commenced at the time of diagnosis (T0), continued with a control visit at 2 months (T2), and culminated in an exit appointment at 6 months following the completion of medical treatment (T6). People classified with severe APTB showed significantly higher levels of IL-6 (14.7 pg/mL; *p* < 0.05) compared to those with mild APTB (7.7 pg/mL) at T0. The AUCs for the ROC curves and the Matthews correlation coefficient values (MCC) demonstrate correlations ranging from moderate to very strong. We conducted WGS on 88 clinical isolates of Mtb, and our analysis revealed a total of 325 genes with insertions and deletions (Indels) within their coding regions when compared to the Mtb H37Rv reference genome. The pattern of association was found between serum levels of CHIT1 and the presence of Indels in Mtb isolates from patients with severe APTB. A key finding in our study was the high levels of CHIT1 in severe APTB patients. We identified a biomarker profile (IL-6, IFN-γ, IL-33, and CHIT1) that allows us to identify individuals with severe APTB, as well as the identification of a panel of polymorphisms (125) in clinical isolates of Mtb from individuals with severe APTB. Integrating these findings into a predictive model of severity would show promise for the management of APTB patients in the future, to guide host-directed therapy and reduce the prevalence of PTLD.

## 1. Introduction

Tuberculosis (TB) is an infectious disease caused by the pathogenic bacterium *Mycobacterium tuberculosis* (Mtb) [1,2,3,4,5]. TB is one of the most important causes of death worldwide, according to the World Health Organization (WHO), with an estimated 10 million cases and 1.5 million deaths worldwide [6]. Active pulmonary TB (APTB) can range from mild disease to severe life-threatening disease [7,8,9]. Early severity classification of APTB would be essential to set up an adequate treatment to prevent the progression of the disease and to reduce its sequelae [10,11]. The severity of APTB in people with a competent immune system and without underlying health conditions may depend on three main factors: the Mtb strain that causes the infection, the host immune response, and the extent of the pathology.

Mtb is a highly diverse pathogen, with multiple variants classified in lineages and sublineages that differ in their geographical distribution, and virulence. The major Mtb lineages include the Euro-American, East Asian, Indo-Oceanic, and African lineages [12]. In recent years, there has been growing interest in the role of Mtb variants in determining the severity of APTB disease [13]. Several studies have shown that Mtb variants are associated with enhanced virulence and differences in APTB severity [14,15,16,17,18,19]. For illustration, the Beijing strain has been associated with increased virulence, drug resistance, and disease severity in multiple studies [20,21,22,23]. Similarly, the Haarlem, LAM, and W-Beijing strains have been associated with increased APTB severity and mortality in some studies [20,24,25,26]. The underlying mechanisms that contribute to these associations are not fully understood but may involve differences in immune evasion, virulence factors, and host–pathogen interactions.

The immune response to Mtb infection is a complex process that involves the production of cytokines by various immune cells, such as macrophages, dendritic cells (DCs), and T cells. Cytokines play a crucial role in the pathogenesis of APTB by regulating the recruitment and activation of immune cells, the formation of granulomas, and the control of bacterial growth. The different Mtb lineages and sublineages differ in their virulence and immunogenicity [27]. For example, some modern Beijing and LAM strains have been associated with lower levels of pro-inflammatory cytokines, such as TNF-α, IL-1β, and IL-6, compared to other Mtb strains that induce higher levels of these cytokines [28,29,30,31]. These immunogenicity differences suggest that Mtb variants may modulate the host immune response by regulating the host production of cytokines [27,32]. Understanding the association between Mtb variants and the production of cytokines during APTB is crucial for the patient’s severity classification. 

Finally, the complex interaction between the host immune response and the pathogen plays a critical role in determining the extent of the Mtb pathology. This extended pathology in the most severe APTB patients can have various sequelae, which are adverse outcomes at the end of treatment [7,33]. For instance, if the infection is localized to a small area of the lung, it may be less severe than if it has spread to multiple areas of the lungs and with the presence of cavities. This damage may be permanent and can lead to long-term health complications. APTB can cause scarring and fibrosis in the lungs, which can make it difficult to breathe and reduce lung function [34]. This can lead to long-term respiratory problems such as chronic obstructive pulmonary disease (COPD) [35]. Close monitoring and follow-up care can help to identify and address any long-term complications of APTB treatment. These sequelae can have substantial consequences on the health and quality of life of APTB patients, and therefore it might be critical to identify biomarkers to prevent these effects on the patients.

In addition to cytokines, the immune response to Mtb infection involves a complex interplay of inflammatory mediators. These mediators play a crucial role in the recruitment and activation of immune cells, such as macrophages and T cells, to the site of infection. One of the rarely reported inflammatory mediators involved in APTB is Chitinase (also known as chitotriosidase or CHIT1), which is produced by a variety of cells in the body, including macrophages, which play a key role in the immune response and pathogenesis during APTB [36]. CHIT1 has been investigated as a potential biomarker for APTB severity. Some studies have shown that CHIT1 levels are elevated in patients with active APTB disease, particularly in those with more severe disease [37,38]. Other studies have suggested that CHIT1 levels may be useful for monitoring disease progression and treatment response in patients with APTB [39]. For example, one study found that CHIT1 levels decreased significantly after six months of APTB treatment, suggesting that it could be a useful marker of treatment response [40].

In this paper, we propose a predictive model for individuals with APTB that includes IL-6 as a severity marker at the time of diagnosis. We identified 125 polymorphisms in Mtb isolates from individuals with a clinically more severe phenotype, associated with high serum levels of CHIT1 after eight weeks of treatment. At two months of treatment, we identified four severity biomarkers (IL-6, IFN-γ, IL-33, and CHIT1) and at the end of treatment, we found five significant biomarkers (MCP-1, IL-18, IL-6, IFN-γ, and CHIT1). In a future study, these biomarkers should be validated for practical application.

## 2. Methods

### 2.1. Ethics Statement

The research was reviewed and approved by the Bioethics Committee of the Faculty of Medicine, University of Antioquia. To ensure confidentiality, each case was anonymized by assigning an identification code. The participants enrolled in the study belonged to the TB Control Program of two Health Benefit Plan Administrators (EPS) in the Metropolitan Area of Medellín, Colombia. All patients signed an approved informed consent approved by the ethics committee of University of Antioquia. The experiments were conducted by the principles expressed in the Declaration of Helsinki.

### 2.2. Study Population

A prospective longitudinal cohort was established, including 108 cases of confirmed active pulmonary tuberculosis (APTB), selected within the first 2 weeks of anti-tuberculosis treatment from December 2020 to July 2022, with a clinical and laboratory follow-up of 6 months. Individuals living with HIV, those receiving immunosuppressive medications, or having comorbidities with other diseases that could affect the immune response (malignant neoplasms, autoimmune diseases, diabetes mellitus, chronic kidney disease), individuals in conditions of homelessness, pharmaco-resistant as evidenced in the antibiogram of the admission culture, and/or mutations associated with antibiotic resistance obtained from the complete genomes of Mtb strains were excluded. In this research, we initiated a follow-up to monitor the treatment progress of 121 cases of APTB. The follow-up commenced at the time of diagnosis (T0), extended with a control visit at 2 months (T2), and concluded with an exit appointment 6 months after the completion of medical treatment (T6).

### 2.3. TB Score

For the clinical evaluation of individuals with APTB included in the study, an adaptation of a previously proposed scoring system [41] was implemented. This instrument includes signs and symptoms such as cough, night sweats, chest pain, dyspnea, hemoptysis, tachycardia (heart rate > 90 beats per minute), axillary temperature > 37 °C, conjunctival pallor, positive findings on auscultation (rhonchi and crepitations), body mass index (BMI) determined by the ratio of weight (kg) to height in meters squared (m^2^), and the circumference of the mid-upper arm on the non-dominant side (MUAC, mm). A value of 1 point is assigned to all signs and symptoms, except for hemoptysis, which is assigned 3 points. An IMC < 18 and MUAC < 220 are given a value of 2 points, resulting in a total score of 15 points. Based on this score, the clinical severity of TB is divided into three classes: Class I (0–4 points), Class II (5–8 points), and Class III (≥9 points) (Appendix A).

### 2.4. Clinical and Radiological Severity Classification

The severity of the disease was quantified at the start of treatment. First, digital chest X-ray images were interpreted using a decision tree adapted from previous studies [8,42], blindly by two independent physicians (a pulmonologist and a radiologist), based on the extent of lesions, presence of pulmonary cavitation, and the size of pleural effusion. Minimal lesions were cases of single hilar enlargement or infiltrates occupying less than one-third of a lung. Moderate lesions had cavities up to 4 cm or pleural effusion extending less than 2 cm from the carina, or lesions of no greater extent than the total volume equivalent to one lung. Advanced lesions exceeded the criteria for moderate. To have a broader view of the clinical severity of the disease, a classification flowchart (Figure 1A) was developed, where the findings from the chest X-ray were combined with the modified TB score.

### 2.5. Recovery of Clinical Mtb Isolates

Mtb was isolated from the sputum of (*n* = 88) individuals with APTB. The decontaminated sputum was cultured on Ogawa Kudoh medium and then transferred to 10 mL of 7H9 liquid medium supplemented with 10% oleic acid-albumin-dextrose-catalase (OADC) growth supplement, 0.05% tyloxapol, and 0.05% glycerol. The samples were verified using the SD BIOLINE TB Ag MPT64 rapid test. Subsequently, the bacteria were cultured at an OD_600_ nm of 0.5, collected by centrifugation (3000 rpm), and the pellet was stored at −20 °C.

### 2.6. Whole-Genome Sequencing and Analysis

The Mtb genomes were sequenced by Novogene (Sacramento, CA, USA) using the Illumina Novaseq 6000 platform, analyzed, and deposited in the NCBI SRA database under the bioproject PRJNA867148 [43].

### 2.7. Cytokine Detection

Cytokines and the enzyme were detected in the serum of individuals with APTB using an immunoassay. This included LEGENDplex™ (BioLegend, San Diego, CA, USA) for IL-1β, IFN-α2, IFN-γ, TNF-α, MCP-1 (CCL2), IL-6, IL-8 (CXCL8), IL-10, IL-12p70, IL-17A, IL-18, IL-23, and IL-33. Serum concentrations of CHIT1 were determined by ELISA (Invitrogen|Thermo Fisher Scientific, Waltham, MA, USA). The values for IL-12p70 and IL-17A were found to be below the detection limit and therefore were not analyzed.

### 2.8. Statistical Analysis

The data were analyzed using GraphPad Prism software, version 8.1.0. The reference to “*n*” is found in the figure legends, and data normality was verified. Median values with interquartile ranges (IQR) were used as measures of central tendency and dispersion. Cytokine levels were compared between two study groups using the Mann–Whitney U test, or the Kruskal–Wallis test for more than 2 groups, followed by post-hoc multiple comparisons tests as indicated in the figure legends. Other statistical tests included the Spearman correlation and chi-square test. Venn diagrams were generated using Venny 2.0.2 BioinfoGP. The area under the ROC curve and the Youden index were determined to indicate the optimal cutoff point. For the Matthews correlation coefficient (MCC), we used the MCC calculator (https://www.mdapp.co/matthews-correlation-coefficient-calculator-374/, accessed on 25 October 2023).The STRING database, Version 11.5 (STRING CONSORTIUM 2022 SIB, CPR, and EMBL; https://string-db.org, accessed on 25 October 2023) was used [44]. All statistical analyses were considered significant at a *p*-value < 0.05.

## 3. Results

### 3.1. Characterization of the APTB Cohort in Antioquia, Colombia

To investigate whether the proposed determinants associated with Mtb could contribute to the severity of APTB, we conducted a prospective cohort study between December 2020 and July 2022. A total of 121 APTB cases were included based on a positive report from serial bacilloscopy in the healthcare centers of the Metropolitan Area of Medellin (Appendix A). Six cases were excluded (two extra-pulmonary TB cases and four cases of antibiotic resistance) based on confirmatory culture results and medical history. Additionally, seven cases with missing general information were excluded from the study. Follow-up started at the time of diagnosis (T0), followed by a control visit at 2 months (T2), and finally an exit appointment at 6 months after completing medical treatment (T6). The general clinical characteristics of the study cohort are presented in Table 1. The median age of the study population was 26.5 years, and the frequency of women was 53.7%. Regarding clinical factors, the median BMI was 19.7 kg/m^2^, and according to the classification of the World Health Organization [6], 36.1% of the patients were underweight, while the majority (63.8%) had a normal weight or were overweight. The median delay in diagnosis, defined as the time from the self-reported onset of typical TB symptoms to the start of anti-tuberculosis treatment [45,46], was 17.4 weeks with a wide interquartile range.

### 3.2. Severity Classification for Individuals with APTB Using a Clinical-Radiological Algorithm at the Time of Diagnosis

Toward our objective, based on chest X-ray classification and clinical parameters, a clinical-radiological algorithm was used to assign the severity of APTB at the time of diagnosis (Figure 1A, Appendix A) and was applied to the 108 APTB patients. Based on this classification, the most common presentation of APTB was moderate (46.3%), followed by severe (29.63%) and mild (24.0%) (Figure 1B). There was a significantly higher proportion of men in the severe APTB group (Figure 1F). However, delay in diagnosis, age, and weight were distributed with no significant differences across the three severity groups (Figure 1C–F).

### 3.3. A Serum Bio-Profile Is Associated with Disease Severity in Individuals with APTB

After classifying individuals in our study cohort according to APTB severity, we compared serum levels of the aforementioned cytokine panel and the CHIT1 protein, to identify biomarkers associated with disease severity at different follow-up times. People classified with severe APTB showed significantly higher levels of IL-6 (14.7 pg/mL; *p* < 0.05) compared to those with mild APTB (7.7 pg/mL) at T0. Additionally, patients with severe APTB exhibited significantly higher levels of IL-6 (8.16 pg/mL; *p* < 0.05), IFN-γ (10.15 pg/mL; *p* < 0.05), IL-33 (118.4 pg/mL; *p* < 0.05), and CHIT1 (23.96 ng/mL; *p* < 0.05) compared to those with mild TB (IL-6: 3 pg/mL; IFN-γ: 4.8 pg/mL; IL-33: 56.89 pg/mL; CHIT1: 11.04 ng/mL) at T2 (Figure 2A,B).

Levels of IL-1β (7.1 pg/mL, *p* < 0.005), IL-8 (11.9 pg/mL, *p* < 0.05), IL-10 (8.6 pg/mL, *p* < 0.005), and IL-23 (9.2 pg/mL, *p* < 0.005) were also significantly higher in patients with severe APTB compared to those with moderate APTB (IL-1β: 4.57 pg/mL, IL-8: 4.77 pg/mL, IL-10: 3.79 pg/mL, IL-23: 3.6 pg/mL) at T2 (Figure 2C). Moreover, no differences were found in cytokine levels between individuals with mild and moderate APTB.

Due that plasma levels of IL-6, IFN-γ, IL-33, and CHIT1 appear to differentiate severe TB from mild APTB during the first 2 months of anti-tuberculosis treatment, we conducted an analysis using ROC curves to determine the diagnostic accuracy of these molecules, concerning APTB severity. The AUCs for the ROC curves were 0.82 (IL-6), 0.79 (IFN-γ), 0.81 (IL-33), and 0.75 (CHIT1) (Figure 2D, Appendix A). As some authors have proposed that the Matthews correlation coefficient (MCC) offers greater reliability and informativeness regarding the accuracy of binary classifications compared to ROC AUC curves, we calculated MCC values for the same analytes: 0.389 (IL-6), 0.523 (IFN-γ), 0.617 (IL-33), and 0.523 (CHIT1). In summary, these ROC and MCC values exhibit correlations that span from moderate to very strong.

### 3.4. Individuals with Severe APTB Exhibit Elevated Levels of Inflammatory Cytokines That Correlate with Regulatory Cytokines

We aimed to examine the correlations among the measured cytokines for all APTB patients after 8 weeks of treatment, using the Spearman rank correlation test. We found significant correlations between different cytokines with r > 0.78 and *p* < 0.0001 (Figure 3). Interestingly, the strongest correlation was found between the regulatory cytokine IL-10 and the inflammatory cytokine IFN-γ (r = 0.91, *p* < 0.0001) (Figure 3). Another significant correlation was found between the inflammatory cytokine IL-23 and the regulatory cytokine IL-33 (r = 0.88, *p* < 0.0001) (Figure 3). It could be possible that severity in APTB patients may be due in part to the imbalance between the inflammatory response (represented here by IFN-γ and IL-23) and the immune regulation (represented here by IL-10 and IL-33).

### 3.5. Pro-Inflammatory Biomarkers at the End of Treatment

Cytokines, chemokines, and inflammation-associated molecules can serve as biomarkers for monitoring treatment response and efficacy in patients with APTB. To understand the behavior in response to anti-tuberculosis treatment, we applied the Wilcoxon rank-sum test and compared the serum levels of the aforementioned analytes at T0 and T6. Our findings showed that serum levels of IL-6 (3.85 pg/mL; *p* < 0.0005), IFN-γ (6.16 pg/mL; *p* < 0.05), CHIT1 (10.75 ng/mL; *p* < 0.005), and IL-18 (213.3 pg/mL; *p* < 0.05) decreased at the end of chemotherapy compared to T0 (IL-6: 10.62 pg/mL; IFN-γ: 9.49 pg/mL; CHIT1: 15.08 ng/mL; IL-18: 303 pg/mL). On the other hand, there was an increase in serum levels of MCP-1 at 6 months (279.7 pg/mL; *p* < 0.0001) of treatment compared to baseline (T0: 129.5 pg/mL) (Figure 4).

### 3.6. Genes Associated with Polymorphisms Present in Clinical Isolates of Mtb from Patients with Severe APTB

While monitoring the patients at T0, we recovered the Mtb bacillus from sputum samples for culture and subsequent DNA extraction. After the complete sequencing of 81 genomes, 4 were excluded due to a high frequency of alternate single nucleotide polymorphisms (AltSNPs) [43]. Lineage and sublineage were predicted using the TB-Profiler web platform (https://tbdr.lshtm.ac.uk/, accessed on 25 October 2023). Notably, all isolates were classified within the MTBC lineage, L4 [43]. Among the sublineage classification, L4.1.2.1 was the predominant sublineage with a frequency of 44.1%, followed by the sublineage L4.3.3 with a frequency of 32.4% (Figure 5A). No sublineage predominated based on the severity group in our APTB cohort, as it has been reported before in other populations (Figure 5B) [13].

To search for bacterial genomic determinants that could explain differential cytokine responses in patients with severe APTB, we next performed an initial analysis to find Mtb genome differences between the groups with extreme severities (mild and severe APTB). We found a total of 325 genes with insertions and deletions (Indels) in coding regions compared to the Mtb H37Rv reference genome. Of the total genes with deletions mapped (*n* = 204), 96 were common in Mtb genes from patients with mild and severe APTB, while 19 were unique to patients with mild APTB and 89 to patients with severe APTB (Figure 5C). Regarding the total genes with insertions mapped (*n* = 121), 66 were common in isolates from patients with mild and severe APTB, while 19 were unique to patients with mild APTB, and 36 to patients with severe APTB (Figure 5C).

### 3.7. CHIT1 Serum Levels Are Associated with a Pattern of Mtb Indels According to Each APTB Patient’s Severity

To determine if there is an association between the presence/absence of exclusive Indels associated with a form of disease severity and serum analyte concentrations, we created a concordance table of serum levels of measured analytes organized from lowest to highest concentration at follow-up times. To differentiate between low and high levels, we used the threshold value indicated by the Youden index. We then compared these concentrations with the presence or absence of genes (Appendix A) with that of polymorphisms in Mtb isolates from patients with mild or severe APTB. To generate the gene list and understand their function, we excluded intergenic regions and repetitive regions, and for the *ppe* and *pe_pgrs* genes, we retained only one variant in case multiple polymorphisms were present in the same gene. A similar analysis was previously conducted by Sousa et al., 2020 [8]. No clear pattern of association was found between APTB severity, serum cytokine levels, and polymorphisms at T0 and T6. However, a pattern of association was found between serum levels of CHIT1 and the presence of Indels in Mtb isolates from patients with severe APTB (Figure 6). Based on the cutoff point, two groups of enriched polymorphisms were found. The first group, located in the lower-left corner of Figure 6, corresponds to genes with polymorphisms associated with severe APTB and high serum levels of CHIT1 (*n* = 95). The other group associated 33 genes with polymorphisms in individuals with mild TB and low serum levels of CHIT1. Quantitative analysis revealed significant differences between the groups of genes with polymorphisms and serum levels of CHIT1 (Fisher exact test, *p* < 0.0001 and OR = 10.89).

Figure 7 shows the genes associated with polymorphisms present in clinical isolates of Mtb from patients with severe APTB. Three main regions were identified (R1, R2, and R3). In the first region (R1), we found polymorphisms present in a two-component regulatory system KdpD/KdpE involved in the regulation of the *kdp* operon (*rv1028c*) and other genes related to a potassium transport ATPase (*kdpB-kdpA*). This two-component system responds to environmental signals such as low potassium ion levels, osmotic imbalance, acid stress, and nutrient availability [47]. Additionally, a polymorphism was found in the Rv3200c protein, a putative potassium channel [48]. The second region (R2) corresponds to the PPE and PE_PGRS protein family members. Here, we highlight a polymorphism in the *pe_pgrs30* gene, which is involved in the suppression of the pro-inflammatory immune response in macrophages through modulation of the host cytokine response and is also associated with mycobacterial latency [49].

Furthermore, a polymorphism was found in the gene encoding the channel protein with necrosis-inducing toxin Rv3903c (CpnT), which plays a dual role in nutrient uptake and the induction of host cell death. Both functions are necessary for the survival, replication, and cytotoxicity of Mtb in macrophages [50]. Other polymorphisms were present in *ppe54*, *pe_pgrs44*, and *rv0538*. In the third region, polymorphisms were found in genes encoding proteins involved in the mycolic acid biosynthesis pathway (monooxygenase-*mymA*, thioesterase-lipR, dehydrogenase-*sadH*, acyl coA synthase-*rv3087*, *rv3088*, and *fadD13*) [51].

In the third region (R3), there are polymorphisms in genes encoding proteins involved in the biosynthesis of mycolic acids (monooxygenase—mymA, thioesterase—lipR, dehydrogenase—sadH, acyl coA synthase—rv3087, rv3088, and fadD13) [52]. Additionally, the gene encoding the MymA protein is essential to maintain the permeability of the mycobacterial envelope when exposed to an acidic pH [51,53]. Mtb FadD13, a fatty acyl-CoA synthetase, presents itself as a promising drug target. This is primarily because its deletion has been demonstrated to significantly decrease the production of pro-inflammatory cytokines, namely IL-1β, IL-18, and IL-6. Moreover, it has been established that Mtb FadD13 is vital for the proliferation of Mycobacterium tuberculosis within macrophages, and it assumes a pivotal role in the generation of pro-inflammatory cytokines during Mtb infection [54]. Also, in this R3 cluster we found the Rv3083 protein, which acts as an agonist of TLR2 and activates macrophages to produce a Th1 immune response [55]. Interestingly, we also observed a polymorphism in the rpfC gene involved in the reactivation of latent mycobacteria [56].

Appendix A presents the functional enrichment analysis for genes associated with polymorphisms present in clinical isolates of Mtb, from patients with severe TB, showing a large network highlighting Gene Ontology (GO) categories based on strength and the false discovery rate (FDR). For molecular function, the activity of K^+^ transporter ATPase was observed (FDR = 1.45 × 10^−5^), while in biological processes, the following was evident: (1) Response to acidic pH (FDR = 2.13 × 10^−11^); (2) Triglyceride biosynthetic process (FDR = 0.0482); (3) Response to nitric oxide (FDR = 0.0482); (4) Glycerol metabolic process (FDR = 0.0482). Regarding the cellular component, the following was shown: (1) Integral component of the plasma membrane (FDR = 0.0371) and (2) Integral component of the membrane (FDR = 0.0371). Additionally, among the UniProt Keywords, potassium transport stands out (FDR = 1.30 × 10^−5^). Within the KEGG pathways, the two-component system is highlighted (FDR = 1.58 × 10^−5^). Finally, describing the protein domains and features (InterPRO), the PE-PGRS family, N-terminal, was found (FDR = 0.0192).

## 4. Discussion

Current guidelines for the clinical care of patients with active pulmonary tuberculosis (APTB) worldwide do not implement strategies for classifying disease severity [6]. Here, we propose a possible predictive model of severity for medical care and follow-up in clinical trials, which provides a set of biomarkers to facilitate decision-making regarding the management, prognosis, and treatment of individuals with APTB (Figure 8).

First of all, our flow cytometry analysis demonstrated that individuals with clinically and radiologically more severe APTB have elevated concentrations of IL-6 in their serum at the beginning of the treatment (T0). This finding is clinically relevant as it could become a tool for identifying patients at a high risk of developing post-tuberculosis lung disease (PTLD). In a previous cohort study conducted in Hong Kong involving individuals with APTB, serum levels of IL-6 were positively correlated with the mycobacterial burden, degree of radiographic consolidation, duration of fever, length of hospitalization, and a clinical severity score for TB [57]. IL-6 was also found to be significantly more elevated in the severe APTB group after 2 months of treatment initiation, together with IFN-γ, IL-33, and CHIT1. In the management of individuals with APTB, serial sputum smear microscopy and culture are the tests used in the second month of treatment to decide whether to transition from an intensive phase to a continuation phase [58]. However, the sensitivity of smear microscopy is limited, and culture has a poor individual predictive capacity to make decisions regarding shortening the treatment [58,59]. Therefore, complementary tools are needed for clinical decision-making at two months, and the identified biomarker profile (IL-6, IFN-γ, IL-33, and CHIT1) could allow for the evaluation of individuals with APTB to determine if they are responding to the antibiotic therapy. These panels of cytokines have been associated with Mtb pathogenesis and may be related to the inability of the cellular immune response to eradicate Mtb. For instance, it has been shown that IL-6 secreted by Mtb-infected macrophages inhibits the responses of uninfected macrophages to IFN-γ [60]. Additionally, IL-6 affects dendritic cell differentiation, inhibiting their antigen-presentation capacity [61]. As for IFN-γ, elevated levels of this cytokine in the broncho-alveolar lavage fluid from APTB patients correlate with disease severity, suggesting a pro-pathological role during the chronic stage of the disease [62]. Moreover, IFN-γ stimulation of Mtb-infected macrophages increases in cells with membrane damage, depending on the mycobacterial strain and macrophage population [31]. IL-33 has not been previously associated with APTB severity. However, during the immune response in TB, IL-33 increases the number of T_reg_ cells and stimulates the production of cytokines such as IL-4, IL-5, and IL-10, as well as the suppression of IFN-γ in polarized Th1 cells, which is permissive for the mycobacteria and detrimental to the host [63,64,65].

We found a correlation between the inflammatory cytokines IFN-γ and IL-23 with the regulatory cytokines IL-10 and IL-33. It has been shown that severity in APTB patients may be due in part to the imbalance between inflammation and regulation. On one hand, while inflammation is essential for clearing an Mtb infection, excessive or prolonged inflammation can lead to lung tissue damage and various pathological conditions in the more severe APTB patients. On the other hand, regulatory responses are beneficial, especially in limiting tissue damage caused by the inflammatory response. However, an excessive regulatory response can also have negative consequences. Elevated levels of IL-10 have been observed in individuals with active TB and have been linked to disease progression, increased bacterial load, and impaired control of the infection. High levels of IL-10 are thought to dampen the pro-inflammatory response necessary for effectively containing and eliminating Mtb. Moreover, in some cases, excessive IFN-gamma production may also contribute to tissue damage and TB severity. Studies have shown that IL-23 is upregulated during active TB disease, and its production correlates with disease severity. IL-23 is another pro-inflammatory cytokine that plays a critical role in the immune response against TB. It is mainly produced by macrophages and dendritic cells and promotes the differentiation of T-helper 17 (Th17) cells. Th17 cells are involved in the recruitment of other immune cells to the site of infection and contribute to granuloma formation, a hallmark of TB infection. The presence of IL-23 in the lung tissues of TB patients indicates its involvement in the immune response at the site of infection. However, excessive or uncontrolled IL-23 production can also contribute to tissue damage and exacerbate inflammation. IL-33 is a member of the IL-1 cytokine family and is known for its role in promoting Th2-type immune responses and contributing to allergic inflammation. However, emerging research indicates that IL-33 may also play a role in the immune response to infectious diseases, including TB. Some studies have shown that IL-33 levels are elevated in the serum and lung tissues of TB patients compared to healthy individuals. Additionally, IL-33 expression is associated with the severity of lung inflammation and tissue damage in TB. The precise mechanisms through which IL-33 influences the immune response to TB and disease severity are not yet fully understood.

A key finding in our study was the high levels of CHIT1 in severe APTB patients. The association between the severity of active pulmonary tuberculosis (APTB) and elevated serum levels of CHIT1 has been previously demonstrated in clinical studies [38,39]. The role in the immunopathology of TB is starting to be unveiled for CHIT1. One of the immunological effects induced by CHIT1 is pulmonary fibrosis through increased TGF-β signaling [66]. It also stimulates various inflammatory mediators such as IL-8, MMP9, MCP-1, RANTES, and eotaxin, inducing cellular migration [67,68]. CHIT1 could be an ideal biomarker because it is one of the most abundant analytes in human lungs and circulation [69,70], as also evidenced in this study. In this study, we proposed that these four biomarkers, identified at T2, would allow for evaluating treatment response and identifying a subgroup of individuals with severe APTB who may develop post-tuberculosis lung disease (PTLD). This approach could lead to promising therapies targeting the host for TB treatment [71,72].

Regarding pro-inflammatory biomarkers at the end of treatment, we found that serum levels of MCP-1 increased while IL-18, IL-6, IFN-γ, and CHIT1 decreased compared to baseline (Time 0). MCP-1 [73], IL-6 [74], IFN-γ [74], and CHIT1 [39] have been reported in similar studies. Regarding IL-18, we did not find reports on its levels at the end of treatment. However, once Mtb is eradicated, there is no stimulus for IL-18 secretion, and its function in inducing IFN-γ production against Mtb is not required [75]. These analytes evaluated at the end of antibiotic therapy could contribute to assessing the restoration of host immune homeostasis, as well as clinical cure and conversion of mycobacterial cultures.

Through whole-genome sequencing (WGS), a group of 125 unique polymorphisms in clinical isolates of Mtb from individuals with severe APTB were found to be associated with high serum levels of CHIT1 after 8 weeks of antibiotic treatment, which had not been reported in the literature before. One possible interpretation is that the heightened virulence of certain Mtb strains might lead to exacerbated inflammation through the production of CHIT1, resulting in lung damage and a more severe form of APTB. Nine of these polymorphisms have been associated with mycobacterial virulence factors. The first one is related to a K^+^ transport complex called KdpFABC and involves a deletion in the two-component system *kdpD*. It was previously described that mice infected with the *kdpDE* mutant died more rapidly than those infected with wild-type bacteria [76]. We also found the presence of four polymorphisms in the genes (*rv3083*, *rv3084*, *rv3085*, and *rv3089*) that compose the *mymA* operon, which is involved in the remodeling of Mtb’s cell envelope under acidic conditions in macrophages [77]. Another polymorphism present in the Mtb isolates from patients with severe TB is in the *pe_pgrs30* gene, and an attenuated phenotype has been observed in mice infected with the *pe_pgrs30* mutant [78]. In vitro, experiments in macrophages confirmed that PE_PGRS30 is required to block phagosome maturation by Mtb [78]. Additionally, two polymorphisms were found in the *rv1787* gene, which encodes the PPE25 protein secreted by ESX-5. PPE25 expressed by *M. smegmatis* induces significantly higher levels of TNF-α and slightly higher levels of IL-1β, mediated by the NF-κB, ERK, and p38 pathways in macrophages [79]. It has also been reported that PPE25 increases the survival of *M. smegmatis* in PMNs, induces cell necrosis, inhibits the expressions of ROS and NO, and alters cytokine secretion, thereby aiding pathogen spread by evading host immunity [80]. Finally, targeted disruption of the *pe_pgrs47* gene (*rv2741*) resulted in attenuated growth of Mtb in vitro and in vivo. The analysis of the effects of *pe_pgrs47* deletion or overexpression implicated this protein in the inhibition of autophagy in infected host phagocytes [81].

The proposed model in this study requires a validation cohort with a much larger sample size and includes different regions of the country. The follow-up of these patients started during the COVID-19 pandemic, which was not considered due to limited knowledge about the disease and the resources it would entail. It would be recommended to use a measurement method with a higher detection capacity in the entire study population. Due to limited knowledge about the functioning of Mtb genes, mechanistic studies are needed to understand the potential effect of the identified polymorphisms, and it would be important to verify the incidence of post-tuberculosis lung disease (PTLD) in the cohort of this study. Furthermore, when considering the implementation of the suggested APTB severity predictive model, it is essential to take into consideration its application in regions with limited resources, particularly in low-income areas. Nevertheless, the potential use of telemedicine and the decreasing cost of laboratory techniques like ELISA for measuring specific analytes may pave the way for its adoption in the near future.

In summary, our research focused on clinical tools, flow cytometry, and WGS. We identified a biomarker bio profile that allows us to classify individuals with severe APTB and identified a panel of polymorphisms in clinical isolates of Mtb from individuals with severe APTB with high serum levels of CHIT1. The integration of these findings into a predictive severity model holds promise for the management of individuals with APTB, aiming to guide host-targeted therapy and reduce the prevalence of PTLD.

## Figures and Tables

**Figure 1 biomedicines-11-03110-f001:**
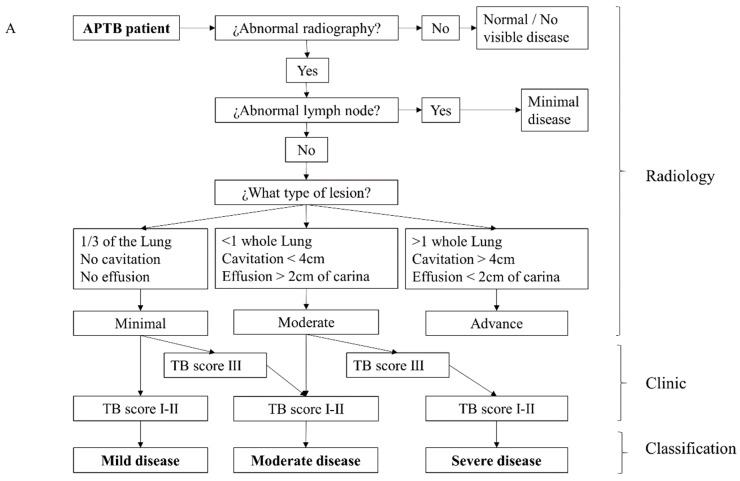
Classification of APTB severity in the study cohort. (**A**) Clinical-radiological algorithm for classifying the severity of APTB at the time of diagnosis as mild, moderate, or severe. (**B**) Patients with APTB for whom we had complete information (*n* = 108) were classified by severity as mild = 26, moderate = 50, and severe = 32. (**C**) Distribution of patient groups by severity and diagnostic delay; (**D**) age; (**E**) weight; and (**F**) gender (male, M; female, F). In (**A**,**F**), frequencies are represented. In (**C**–**E**), the median, maximum value, and minimum value are represented. A Kruskal–Wallis test was used to compare the medians, Dunn’s test for pairwise comparisons between each independent group, and * denotes a *p*-value < 0.05.

**Figure 2 biomedicines-11-03110-f002:**
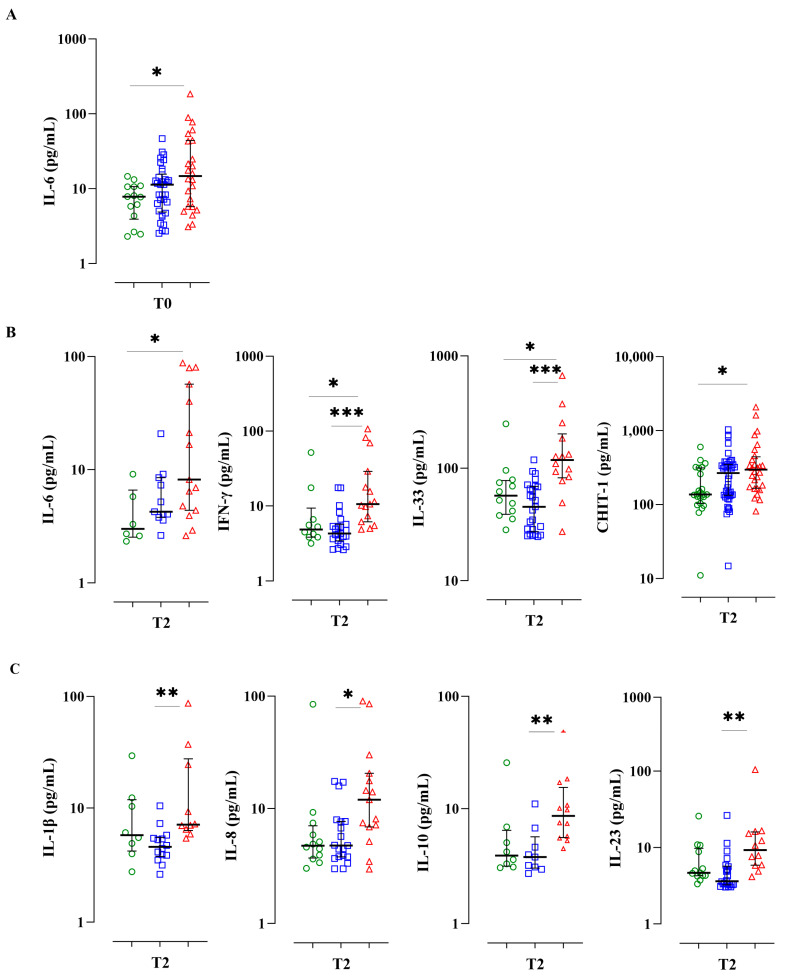
Plasma levels of cytokines IL-6, IFN-γ, IL-33, and the enzyme CHIT-1 can differentiate severe APTB from mild APTB during the first 2 months of anti-tuberculosis treatment. Patients were categorized based on the severity of pulmonary APTB as mild (L) (green), moderate (M) (blue), and severe (S) (red). Serum levels of different analytes were quantified (**A**) at the time of diagnosis: IL-6 (L, *n* = 14; M, *n* = 33; S, *n* = 24), (**B**) at 8 weeks of treatment, showing differences between individuals with mild and severe APTB: IL-6 (L, *n* = 6; M, *n* = 11; S, *n* = 15); IFN-γ (L, *n* = 10; M, *n* = 23; S, *n* = 15); IL-33 (L, *n* = 12; M, *n* = 25; S, *n* = 14); CHIT1 (L, *n* = 23; M, *n* = 44; S, *n* = 28), and (**C**) at 8 weeks of treatment, showing differences between individuals with moderate and severe APTB: IL-1β (L, *n* = 8; M, *n* = 14; S, *n* = 10); IL-8 (L, *n* = 12; M, *n* = 19; S, *n* = 15); IL-10 (L, *n* = 8; M, *n* = 9; S, *n* = 12); and IL-23 (L, *n* = 13; M, *n* = 22; S, *n* = 12). (**D**) ROC curves to estimate the ability of serum biomarkers to distinguish between mild and severe TB. The diagonal line denotes the ROC curve of a random classifier (red) and the actual data ROC curve (blue); ROC, Receiver Operating Characteristic; CHIT1, Chitinase 1. Time of diagnosis (T0); control visit at 2 months (T2); exit appointment at 6 months following the completion of medical treatment (T6). Data represent medians and interquartile ranges. Kruskal-Wallis test and Dunn’s multiple comparisons test were used to compare the distributions of serum cytokines between study groups. Significant *p*-values are presented in the figures (* < 0.05, ** < 0.005 and *** < 0.0005), and the horizontal line represents the groups where the difference was observed.

**Figure 3 biomedicines-11-03110-f003:**
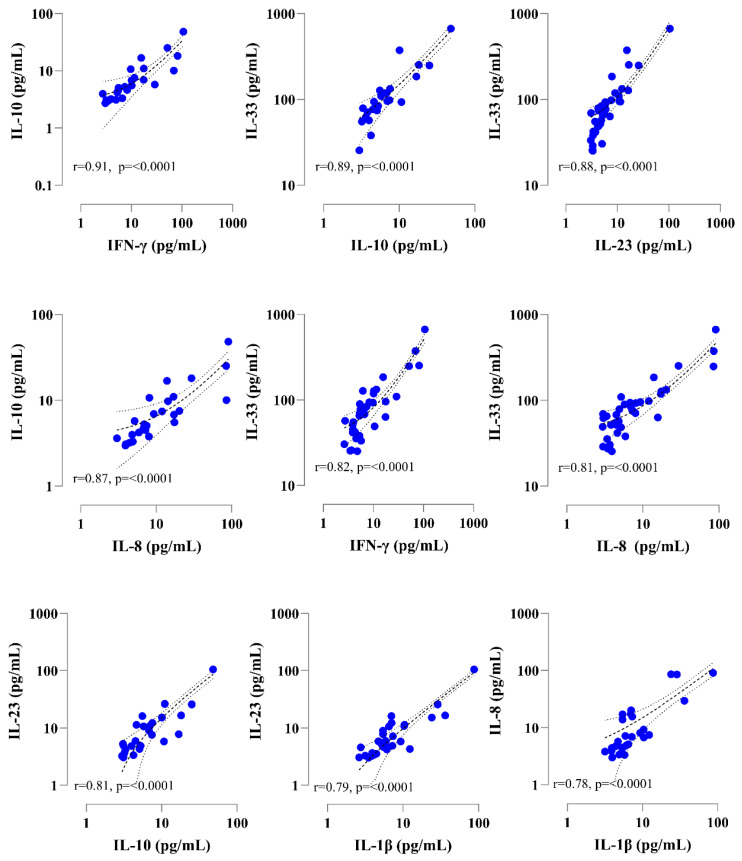
Correlation between serum cytokine levels in patients with severe APTB (advanced pulmonary tuberculosis) at T2 (8 weeks after treatment). The correlations were graphed with the value of statistically significant Spearman correlation coefficient (r). Dashed lines indicate the threshold for statistical significance. The lines with a 95% confidence interval and the curve of the best-fit linear regression are depicted as continuous lines. The *p*-values and R-values are specified in each graph.

**Figure 4 biomedicines-11-03110-f004:**
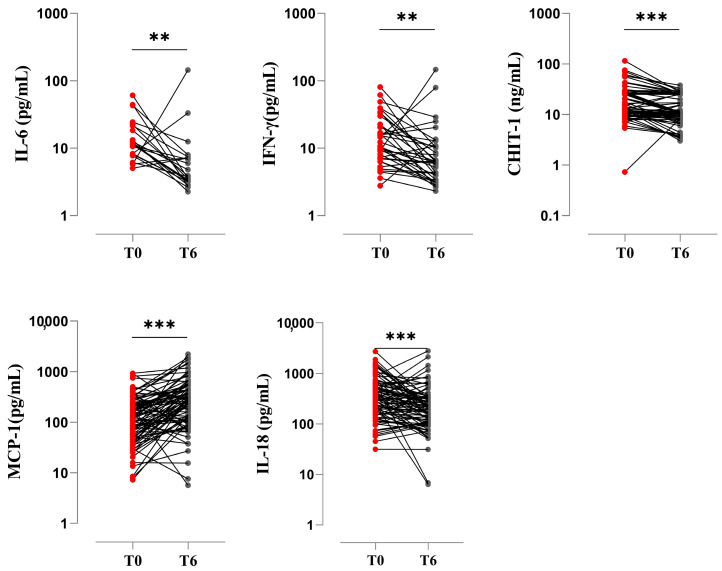
Pro-inflammatory biomarkers at the end of treatment. Serum levels of analytes were compared between individuals with APTB at baseline (*n* = 108) at the end of treatment (*n* = 92). A Wilcoxon signed-rank test with a two-tailed hypothesis test was used. Significant *p*-values are represented in the figures (**, *p* < 0.005; and ***, *p* < 0.0005). Time of diagnosis (T0); control visit at 2 months (T2); exit appointment at 6 months following the completion of medical treatment (T6).

**Figure 5 biomedicines-11-03110-f005:**
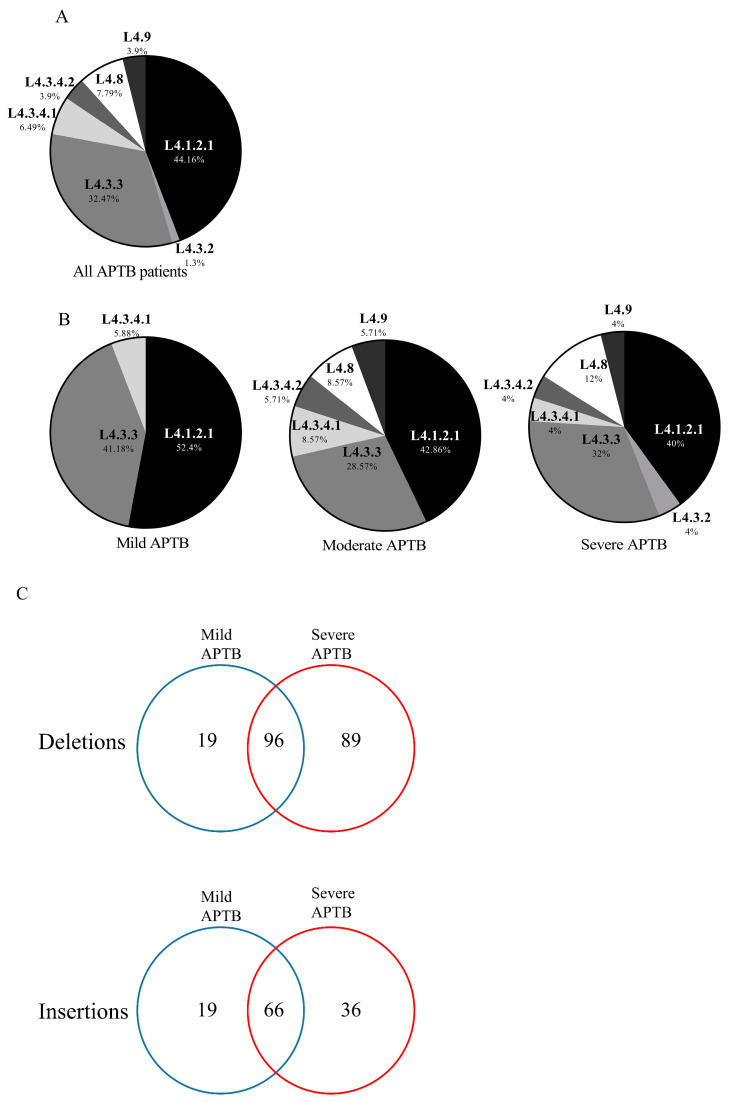
Analysis of clinical Mtb isolates from patients with APTB. The distribution of sublineages assigned to each Mtb isolate using the TB-profiler web server is shown for (**A**,**B**). (**A**) A total of 77 Mtb isolates from patients with APTB were analyzed. (**B**) Mild APTB: 17 Mtb isolates from patients with mild APTB; Moderate TB: 35 Mtb isolates from patients with moderate TB; and Severe TB: 25 APMtb isolates from patients with severe TB. (**C**) For the datasets, intergenic regions and repetitive regions were excluded, and for the *ppe* and *pe_pgrs* genes, only one variant was included in case multiple polymorphisms were present in the same gene.

**Figure 6 biomedicines-11-03110-f006:**
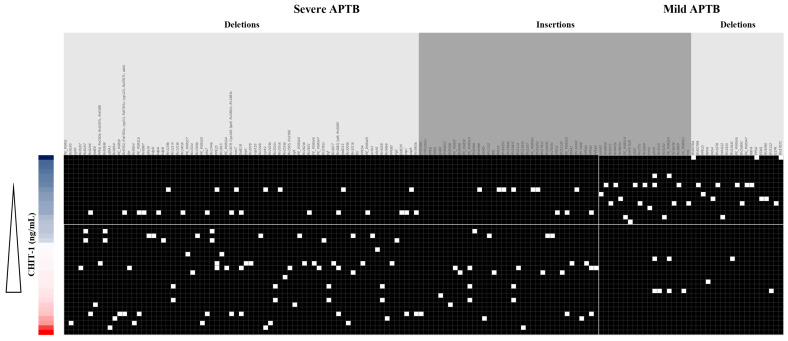
Genes with common polymorphisms in groups of clinical Mtb isolates correlate with elevated serum concentrations of CHIT1 in patients with pulmonary APTB after 2 months of treatment. The cutoff point is 13.25 ng/mL, separating low and high serum levels of CHIT1, calculated using the Youden index. The light gray color represents unique insertions per severity group, and the dark gray color represents unique deletions per severity group. The left side of the figure represents polymorphisms found in clinical Mtb isolates from patients with severe APTB, and the right side represents polymorphisms found in clinical Mtb isolates from patients with mild APTB. Genes with (white squares) or without (black squares) polymorphisms are indicated.

**Figure 7 biomedicines-11-03110-f007:**
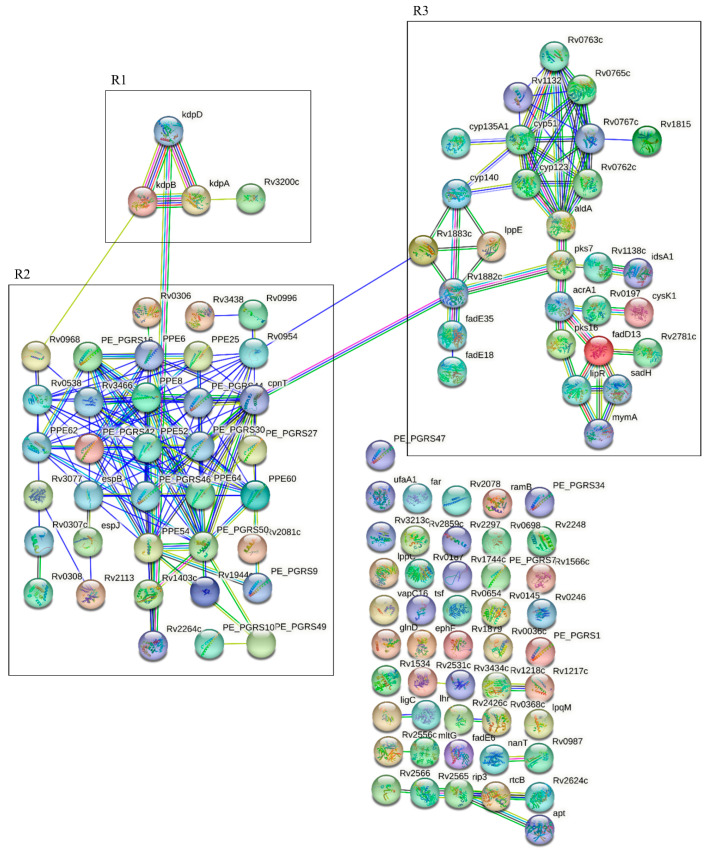
Mtb Indels STRING analysis. Functional analysis of genes associated with polymorphisms present in clinical Mtb isolates from individuals with mild and severe APTB. In the diagram, we have categorized three primary clusters as R1, R2, and R3. R1 represents the KdpDE two-component system, R2 corresponds to members of the PPE and PE_PGRS protein families, and R3 primarily consists of members from the mycolic and fatty acid synthesis.

**Figure 8 biomedicines-11-03110-f008:**
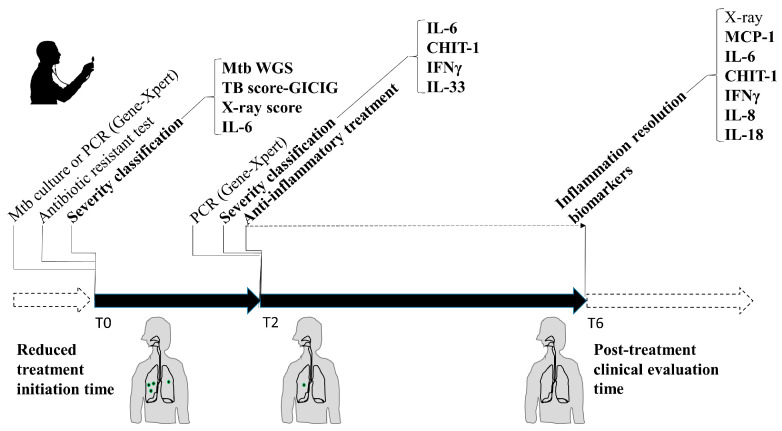
Treatment and diagnostic model for APTB including severity biomarkers. At T0, a severity stratification should be included based on the TB score, X-ray, serum IL-6 levels, and Mtb WGS. Additionally, the bacilloscopy should be replaced by a PCR test like Gene Xpert. At T2, IL-6, CHIT1, IFNγ, and IL-33 levels should be measured and severity should be determined by the levels of these biomarkers. If the levels correspond to a severe APTB it could be important to initiate a complementary anti-inflammatory treatment. Finally, at T6, it would be important to determine the levels for MCP-1, IL-6, CHIT1, IFNγ, IL-8, and IL-18 to establish a minimal inflammatory condition at the end of treatment. If the inflammatory pattern persists, it would be important to continue with a careful follow-up of the patient, to avoid future sequelae.

**Table 1 biomedicines-11-03110-t001:** Demographic and clinical characteristics of APTB patients.

Variable	Descriptor	Cases
**Sociodemographic Factors**		
Age	Median (IQR)	26.5 (22–34.7)
Sex	Male *n* (%)	50 (46.2%)
	Female	58 (53.7%)
**Clinical-Biological Factors**		
Weight (kg)	Median (IQR)	53.9 (49.5–60.4)
Height (m)		1.64 (1.5–1.7)
BMI (kg/m^2^)		19.7 (17.8–22.05)
**BMI Classification (WHO)**	Underweight BMI < 18.5, *n* (%)	39 (36.1%)
	Normal BMI 18.5–24.9	57 (52.7%)
	Overweight BMI 25–29.9	12 (11.1%)
**Previous APTB episode**	No *n* (%)	107 (99.07%)
	Yes	1 (0.9%)
**Contact with APTB patient**	No *n* (%)	94 (87.03%)
	Yes	14 (12.9%)
**Positive radiological findings**	No *n* (%)	2 (1.85%)
	Yes	106 (98.14%)
	Extent of parenchymal involvement	
	<1/3 of one lung *n* (%)	33 (31.1%)
	<1 complete lung	47 (44.3%)
	2 lungs	26 (24.5%)
	Lung cavitation	
	Yes, *n* (%)	38 (35.18%)
	No, *n* (%)	70 (64.81%)
	Pleural effusion	
	Yes, *n* (%)	2 (1.85%)
	No, *n* (%)	106 (98.14%)
**Total diagnostic delay in weeks**	Median (IQR)	17.4 (9.14–41.6)
**Bacilloscopy**	Total samples	99
	1 to 10 AFB in 100 fields *n* (%)	11 (11.1%)
	10 to 99 AFB in 100 fields (1+)	36 (36.36%)
	1 to 10 AFB per field (2+)	27 (27.27%)
	10 AFB per field (3+)	25 (25.25%)
**TB score-GICIG**	SC I 1–4 *n* (%)	33 (30.5%)
	SC II 5–8	51 (47.2%)
	SC III >= 9	24 (22.2%)
**Past Medical History**	Alcohol consumption *n* (%)	10 (9.2%)
	Smoking	9 (8.3%)
	Respiratory	3 (2.7%)

IQR, interquartile range; *n* (%), absolute number and percentage; kg, kilograms; m, meters; BMI, body mass index; kg/m^2^, kilograms per square meter; SC, score; AFB, acid fast bacilli.

## Data Availability

Data are contained within the article.

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
