# Peer review of "Tuberculosis Severity Predictive Model Using Mtb Variants and Serum Biomarkers in a Colombian Cohort of APTB Patients"

_biomedicines, 2023, doi:10.3390/biomedicines11123110_

Round 1

Reviewer 1 Report

Comments and Suggestions for Authors

1. The style of data presentation is not friendly to readers for reading the results. For example, what is the To, T2 and T6 in figure 2 and 4? What are three groups in figure 2C? The figure legend needs to be largely revised.

2. The figure 5-7 could be reorganized into one figure.

3. This study is mor descriptive without mechanistic insights.

4. The legend of figure need to be revised. The information for the experiments design in figure 7 is not clear.

5. The style of references were not consistent, especially in the title of references. Please revised it.

Comments on the Quality of English Language

Minor editing of English language required

Author Response

We added the response in the pdf file

Reviewer 2 Report

Comments and Suggestions for Authors

The study entitled „Tuberculosis severity predictive model using Mtb variants and serum biomarkers in a Colombian cohort of APTB patients” is interesting and innovative. The results are well-organized, and I completely agree with the discussion and conclusions. Moreover, the language used is very clear, making the study easy to read and understand. While I did not find any methodological errors, I do have one suggestion regarding „A serum bio-profile is associated with disease severity in individuals with APTB” section (last paragraph). I suggest to consider providing Matthews correlation coefficient analysis for the classification. Please see the paper:

Chicco, D., Jurman, G. The Matthews correlation coefficient (MCC) should replace the ROC AUC as the standard metric for assessing binary classification. BioData Mining 16, 4 (2023). https://doi.org/10.1186/s13040-023-00322-4

Based on this study, it appears that MCC (Matthews Correlation Coefficient) is a better indicator of classification quality than the ROC curve. I understand that ROC is probably more commonly used, but please look into this matter.

Author Response

We added the response to the reviewer in a pdf file.

Reviewer 3 Report

Comments and Suggestions for Authors

The manuscript entitled "Tuberculosis severity predictive model using Mtb variants and serum biomarkers in a Colombian cohort of APTB patients." by Ocampo and colleagues describe a tuberculosis severity predictive model based on both the analysis of Mycobacterium isolates genomes sequences and serum biomarkers, in a Colombian cohort of patients with active pulmonary tuberculosis. The authors followed a cohort of 108 patients with active pulmonary tuberculosis for 6 months and used the patients blood markers as well as the genome sequences of isolates from patients. The work is well described and written, and the topic is within the scope of the journal.

There are some issues the authors should address:

1) Although the objectives of the study are clearly stated and well defined, I wonder about the applicability of the predictive methodology worldwide, in particular in low income countries. The authors should include in the discussion one or two sentences about the implementation and applicability of the method in low income countries or regions without access to the clinical facilities required.

Abstract: The sentence "The AUCs for the ROC curves were 0.82 (IL-6), 0.79 (IFN-γ), 0.81 (IL33), and 0.75 (CHIT1)." is hard to understand out of the context without any further explanation in the Abstract. In addition, in the next sentence "We found a total of 325 genes with insertions and deletions (Indels) in coding regions compared to the Mtb H37Rv reference genome", no information is provided concerning the number of isolates sequenced. It should also be clarified if the 325 genes present "insertions AND deletions" or "insertions OR deletions".

Page 7: in the abbreviations defenitions on the bottom of table 1, AFB should also be described.

Page 14, Figure 5 legend: The sentence ". The distribution of sublineages assigned to each Mtb isolates using the TB-profiler web server is shown." should appear immediately before "(B) Mild...."

Page 16, Figure 7: Some explanation should be provided concerning the groups R1, R2 and R3.

Page 17, Figure 8: The choice of the color GREEN for part of the written sentences is not a good choice,as when printed in black and white, the green letters are almost unreadable.

References: the references are not in the journal style.

Author Response

(The authors gave the same response as above.)

Round 2

Reviewer 1 Report

Comments and Suggestions for Authors

The authors have addressed my comments.

Reviewer 3 Report

Comments and Suggestions for Authors

The issues raised to the previous version of the manuscript were correctly addressed in this revised version. There are a few minor issues that the authors should address in this revised version:

1) The title should not contain abbreviations. Mtb and APTB should be in full in the title

2) Although a list of abbreviations is provided by reviewers, the first time AUC and ROC appear should be in full

3) In the last part of the results, the authors added novel references with the indication of the respective DOI. These references should be included as regular references.